# Lipid Nanoparticles Loaded with Farnesol or Geraniol to Enhance the Susceptibility of *E. coli* MCR-1 to Colistin

**DOI:** 10.3390/pharmaceutics13111849

**Published:** 2021-11-03

**Authors:** Chantal Valcourt, Julien M. Buyck, Nicolas Grégoire, William Couet, Sandrine Marchand, Frédéric Tewes

**Affiliations:** 1INSERM U1070 “Pharmacology of Anti-Infective Agents”, 1 rue Georges Bonnet, Pôle Biologie Santé, 86022 Poitiers, France; chantal.valcourtsainz@gmail.com (C.V.); julien.buyck@univ-poitiers.fr (J.M.B.); nicolas.gregoire@univ-poitiers.fr (N.G.); william.couet@univ-poitiers.fr (W.C.); sandrine.marchand@univ-poitiers.fr (S.M.); 2UFR Médecine-Pharmacie Université de Poitiers, 6 rue de la Milétrie, TSA 51115, 86073 Poitiers, France; 3Laboratoire de Toxicologie-Pharmacocinétique, CHU de Poitiers, 2 rue de la Miletrie, 86021 Poitiers, France

**Keywords:** lipid nanoparticle, antibiotic-non-antibiotic combination, *mcr*-1, *E. coli*, time-kill curve, E_max_ model, farnesol

## Abstract

Resistance to colistin, one of the antibiotics of last resort against multidrug-resistant Gram-negative bacteria, is increasingly reported. Notably, MCR plasmids discovered in 2015 have now been reported worldwide in humans. To keep this antibiotic of last resort efficient, a way to tackle this mechanism seems essential. Terpene alcohols such as farnesol have been shown to improve the efficacy of some antibiotics. However, their high lipophilicity makes them difficult to use. This problem can be solved by encapsulating them in water-dispersible lipid nanoparticles (LNPs). The aim of this study was to discover, using checkerboard tests and time-kill curve experiments, an association between colistin and farnesol or geraniol loaded in LNPs, which would improve the efficacy of colistin against *E. coli* and, in particular, MCR-1 transconjugants. Then, the effect of the combination on *E. coli* inner membrane permeabilisation was evaluated using propidium iodide (PI) uptake and compared to human red blood cells plasma membrane permeabilisation. Both terpene alcohols were able to restore the susceptibility of *E. coli* J53 MCR-1 to colistin with the same efficacy (E_max_ = 16, i.e., colistin MIC was decreased from 8 to 0.5 mg/L). However, with an EC_50_ of 2.69 mg/L, farnesol was more potent than geraniol (EC_50_ = 39.49 mg/L). Time-kill studies showed a bactericidal effect on MCR-1 transconjugant 6 h after incubation, with no regrowth up to 30 h in the presence of 1 mg/L colistin (1/8 MIC) and 60 mg/L or 200 mg/L farnesol or geraniol, respectively. Colistin alone was more potent in increasing PI uptake rate in the susceptible strain (EC_50_ = 0.86 ± 0.08 mg/L) than in the MCR-1 one (EC_50_ = 7.38 ± 0.85 mg/L). Against the MCR-1 strain, farnesol-loaded LNP at 60 mg/L enhanced the colistin-induced inner membrane permeabilization effect up to 5-fold and also increased its potency as shown by the decrease in its EC_50_ from 7.38 ± 0.85 mg/L to 2.69 ± 0.25 mg/L. Importantly, no hemolysis was observed for LNPs loaded with farnesol or geraniol, alone or in combination with colistin, at the concentrations showing the maximum decrease in colistin MICs. The results presented here indicate that farnesol-loaded LNPs should be studied as combination therapy with colistin to prevent the development of resistance to this antibiotic of last resort.

## 1. Introduction

Antibiotic (ATB) resistance is emerging worldwide, and polymyxins, such as colistin, are among the few ATBs still active against multidrug-resistant (MDR) Gram-negative bacteria such as carbapenemase-producing *Enterobacteriaceae* [1]. Thus, colistin is used as an ATB of last resort to treat infections involving these bacteria [2]. However, colistin-resistant Gram-negative isolates are increasingly being reported. For example, colistin resistance due to chromosomally-encoded modification of LPS of Gram-negative bacteria have spread in some hospitals [3]. Similarly, a series of mobilized colistin resistance (*mcr*) plasmid genes are responsible for the development of polymyxin resistance in several *Enterobacteriaceae*, including *Escherichia coli* and *Klebsiella pneumoniae* [4,5,6,7,8,9,10,11]. Although the resistances given by *mcr*-type genes are generally considered as relatively low, with MIC values for colistin often around 2 to 8 mg/L [8,11,12,13], they can favour the development of other resistance mechanisms such as mutations in the two-component systems PhoPQ and PmrAB, which generally result in higher resistance to colistin (MIC around 8 to 128 mg/L) [14,15,16]. Furthermore, plasmid-borne *mcr* genes can be transmitted via horizontal gene transfer and have the potential to spread globally and rapidly. Indeed, bacteria harboring *mcr*-like gene have been detected in hospital isolates worldwide [17,18,19,20,21], sometimes in multidrug-resistant strains containing *mcr-1* alongside β-lactamase and carbapenemase resistance genes [22]. Therefore, due to the limited number of new ATBs, alternative strategies are needed to prevent the development of these resistances and maintain the efficacy of colistin, which is often the only last-resort therapeutic option.

The combination of ATBs is common in the treatment of severe infections caused by MDR bacteria [12], but the association of ATB with non-ATB is another interesting approach that is, except for β-lactam/β-lactamase inhibitors combinations, still rarely explored. Non-ATB drugs, such as acyclic terpene alcohols, in particular farnesol (FAR) and geraniol (GER) (Figure 1), have shown the potential to enhance ATB efficiency against Gram-positive [23,24,25,26,27,28,29,30] and Gram-negative bacteria [31,32,33,34,35].

For example, FAR potentiates the activity of ampicillin and oxacillin against *Staphylococcus aureus* strains, including methicillin-resistant strains [24,25]. FAR deregulates genes involved in the synthesis of multidrug efflux pumps and cell membrane biogenesis of *A. baumannii*, and acts synergistically with colistin to kill MDR *A. baumannii* [35].

Hence, these terpene alcohols exhibit interactions with ATBs that seem valuable for adjuvant therapy of MDR Gram-negative bacteria, including colistin-resistant isolates. Although many patents on the antimicrobial activity of these terpene alcohols have been granted, their high lipophilicity and low aqueous solubility make them difficult to administer, impeding their development and clinical application. One option to address this issue is to load these molecules into water-dispersible lipid systems such as lipid nanoparticles (LNP). Thus, the aim of this study was to discover, using checkerboard tests and time-kill curve experiments, an association between colistin and FAR- or GER-loaded LNPs that would improve the efficacy of colistin against *E. coli*, particularly *mcr*-1 transconjugants. Then, the effect of the combination on the permeabilisation of the plasma membrane of *E. coli* has been evaluated and compared to the plasma membrane permeabilisation effect of human red blood cells.

## 2. Materials and Methods

### 2.1. Materials

Labrafac^®^ WL 1349 (caprylic/capric acid triglycerides) was provided by Gattefossé S.A. (Saint-Priest, France). Phospholipon HS90 (hydrogenated lecithin) and Solutol HS15 (macrogol 15 hydroxystearate, polyoxyl 15 hydroxystearate; a mixture of free polyethylene glycol 660 and polyethylene glycol 660 hydroxystearate) were kindly provided by Lipoid Gmbh (Ludwigshafen, Germany) and BASF Pharma (Levallois-Perret, France), respectively. Farnesol, geraniol, colistin sulfate (lot: SLBT0851), cation-supplemented Mueller Hinton broth (MHB) and MH agar were purchased from Sigma Aldrich (Saint-Quentin-Fallavier, France).

### 2.2. Bacterial Strains

*E. coli* J53 strain (KACC 16628) susceptible to colistin (MIC = 0.25 mg/L), and its corresponding MCR-1 transconjugant (*E. coli J53* MCR-1) [36] having reduced susceptible to colistin (MIC = 8 mg/L) were used.

### 2.3. Preparation and Characterization of Lipid Nanoparticles (LNPs) Loaded with Terpenic Alcohols

LNPs were prepared following the procedure described by Anton et al. [37]. Briefly, the oil phase of the LNPs made of triglycerides (Labrafac^®^) was mixed under magnetic stirring with water in the presence of NaCl and two surfactants (Solutol HS15 and Phospholipon). This oil-in-water emulsion was heated above the phase-inversion temperature (PIT) (90 °C), to obtain a water-in-oil emulsion. Then, it was cooled down below the PIT at 60 °C, leading back to the formation of an oil-in-water emulsion. Three of those temperature cycles were carried out, and then geraniol or farnesol was added in the last cycle and the system was then quenched by adding 12 mL of cold water (4 °C). The precise amount of each LNP-forming excipient is listed in Table 1. The theoretical concentration of terpene alcohols in the final suspension of nanoparticles was 28.5 g/L.

Size distributions of LNPs were determined by dynamic light scattering measurements using a Malvern NanoZS (Malvern, Orsay, France). LNP suspensions were diluted (1:60 *v*/*v*) with MilliQ water and analyzed in triplicate at 25 °C. LNPs were characterized by the median value of the volume-weighted size distribution and by the polydispersity index (PDI), which is a measure of the broadness of a size distribution.

### 2.4. Evaluation of the Effect of Terpene Alcohol-Loaded LNP on Colistin Efficacy against E. coli–Checkerboard Test

To perform the checkerboard tests, the MICs of colistin against *E. coli* J53 and its transconjuguant MCR-1 were determined in MHB using the microdilution method described in the EUCAST guidelines [38] in the presence of different concentrations of terpene alcohols solubilized in MHB as LNP. Two-fold serial dilutions of colistin in MHB were prepared to obtain final concentrations ranging from 0.06 to 16 mg/L. For each colistin concentration, a two-fold serial dilution of the terpene alcohol-loaded LNP, or equivalent blank particles, was performed in MHB to obtain final concentrations of terpene alcohol ranging from 0.5 to 320 mg/L. Then, the 96-well plates were inoculated with each *E. coli* strain to reach the bacterial concentration of 1 × 10^6^ CFU/mL and incubated for 16–20 h at 37 °C. Experiments were conducted in triplicate on three different days (*n* = 9). An inhibitory E_max_ model (Equation (1)) developed from a study published by Chauzy et al. [39] was used to describe the average decrease in colistin MIC values ($\hat{\mathrm{MIC}}$) in relation with the terpene alcohol concentrations (C_terpene_).
(1)$$\hat{\mathrm{MIC}}={\mathrm{MIC}}_{0}-\frac{\left({\mathrm{MIC}}_{0}-{\mathrm{MIC}}_{\infty }\right)*\;C_{\mathrm{terpene}}^{\gamma }}{{\mathrm{EC}}_{50}{}^{\gamma }+C_{\mathrm{terpenes}}^{\gamma }}$$

In Equation (1), MIC_0_ refers to the mean colistin MIC measured without terpene alcohols. MIC_∞_ refers to the lowest mean colistin MIC measured in the presence of terpene alcohols. The maximal percentage in colistin MIC decrease (E_max_) was calculated as the percentage of MIC_0_/MIC_∞_. EC_50_ is the terpene alcohol concentration (mg/L) producing 50% of the E_max_ and *γ* is the slope factor (Hill coefficient) that measures the sensitivity of the effect within the terpene alcohols concentration range. $\hat{\mathrm{MIC}}$, were evaluated for each terpene alcohol concentration based on the hypothesis that individual MIC values were normally distributed around $\hat{\mathrm{MIC}}$ with a standard deviation σ; $\left(\mathrm{MIC}\;\sim \;N\left(\hat{\mathrm{MIC}},\;\sigma \right)\right)$. Parameters were evaluated using the Rstan software ver. 2.3 by performing Bayesian data analysis and using the following weakly informative prior distributions:$\;{\mathrm{MIC}}_{0}\;\sim \;U\left(0,\;10\right)$; ${\mathrm{EC}}_{50}\;\sim \;half-N\left(0,\;100\right);$
γ ~ half−N(0, 10);
σ ~ half−Cauchy(0, 40). Simulations were run for 4 chains, with 1000 burn-in steps followed by 1000 Markov chain Monte Carlo (MCMC) steps.

### 2.5. Time-Kill Studies

Time-kill studies were performed from fresh *E. coli* J53 and *E. coli J53*_MCR-1 overnight cultures adjusted to an inoculum of 1 × 10^6^ CFU/mL in 15 mL of MHB after 2 h of pre-incubation to obtain logarithmic phase cultures. Colistin was used at concentrations equal to 1/8 of the respective MIC for each strain, i.e., 0.031 mg/L and 1 mg/L for *E. coli J53* and *E. coli* J53_MCR-1, respectively. These colistin concentrations correspond to clinically achievable lung concentrations of colistin in humans using the usual colistin doses. Based on the E_max_ model fitting parameter values obtained in the checkerboard experiments, the concentrations of terpene alcohols tested as LNPs were 10, 30 and 60 mg/L for farnesol and 60, 100 and 200 mg/L for geraniol. The suspensions were incubated at 37 °C under agitation and samples were taken after 0, 3, 6, 24, and 30 h of incubation. The numbers of CFU/mL were determined after making appropriate dilutions of the samples in PBS, and spreading 100 µL on MH agar plates, which were then incubated for 16 h at 37 °C for colony counting. All time-kill curves were performed in triplicate.

### 2.6. Propidium Iodide Uptake-Evaluation of the Permeabilization of the Bacterial Plasma Membrane Induced by Colistin

Propidium iodide (PI) uptake was used to measure inner membrane permeability [40,41]. PI solutions at 40 µM (4×) in PBS (pH 7.4) were prepared from a stock solution at 4 mM in water. Four-time concentrated terpene alcohol-loaded LNP suspensions in PBS were prepared from the stock suspension of LNP. Hence, farnesol concentration was 120 mg/L (4×) and geraniol concentrations was 360 mg/L (4×). Suspensions of blank particles were prepared in the same conditions. From these suspensions, two-times concentrated colistin/terpene alcohol mixes were prepared. *E. coli J53* or the MCR-1 transconjugant were incubated to mid-log phase in MHB at 37 °C, washed twice in PBS, and diluted to obtain OD_600_ = 0.3 in PBS. A 96-well plate was filled with 50 µL of bacterial suspension, then 100 µL of the two-times concentrated colistin/terpene alcohol mix was added to the wells and the plate was placed in an Infinite M200 microplate reader (Tecan^®^, Männedorf, Switzerland). Immediately, 50 µL of the four-times concentrated PI solution was added with the plate dispenser and the fluorescence (excitation λ = 560 nm, emission λ = 630 nm) was recorded every 2 min for 10 min using an Infinite M200 Pro microplate reader (Tecan^®^, France). A fluorescence increase rate was calculated in the initial conditions (linear fluorescence increase with time). PI uptake was calculated as follows: PI uptake = (R_obs_ − R_0_)/(R_100_ − R_0_) where R_obs_ was the rate of increase in fluorescence measured for a given concentration of colistin (*C_COLI_*), R_0_ was the rate of increase in fluorescence observed in the absence of colistin or terpene alcohol, and R_100_ the rate of increase in PI fluorescence measured in the presence of 32 mg/L colistin in the absence of terpene alcohol. Data (3 ≤ *n* ≤ 9) were analyzed using the Rstan software ver. 2.3 by performing Bayesian data analysis and using the following E_max_ model:(2)$$\hat{E}=E_{\min}+\frac{\left(E_{\max}-E_{\min}\right)*\;C_{COLI}^{\gamma }}{{\mathrm{EC}}_{50}{}^{\gamma }+C_{COLI}^{\gamma }}$$

In Equation (2), $\hat{E}$ is the average relative PI fluorescence increase rate. EC_50_ is the colistin concentration (mg/L) producing 50% of E_max_ and *γ* is the slope factor (Hill coefficient) that measures the sensitivity of the effect within the terpene alcohols concentration range. E_max_ and E_min_ are the maximal and minimal relative PI fluorescence increase rate measured, respectively.

### 2.7. Hemolysis Test from Human Erythrocytes

Hemolysis tests were performed to evaluate the possible potentiation of colistin-induced plasma membrane permeabilization of human cells by terpene-loaded LNPs. The protocol used was based on a study by Serrano et al. [42]. Venous blood obtained from healthy volunteers was collected in tubes containing EDTA. Blood was centrifuged for 10 min at 1200× *g* and the supernatant was pipetted off. Red blood cells (RCBs) were then washed twice with isotonic PBS and dispersed in PBS at a cell density of 6.5 × 10^8^ cells/mL. Then, 500 µL of RCB suspension was mixed with 500 µL of PBS containing colistin alone at various concentrations (µg/mL) or supplemented with LNP. LNP were either blank, loaded with FAR to reach a final concentration of 60 mg/L or loaded with GER to reach a final concentration of 100 mg/L. A solution of Triton-X 100 at 1% m/v was used as positive control. Negative control was pure PBS. The RCBs were then incubated for 1 h at 37 °C in a shaking incubator. Unlysed RBCs were removed by centrifugation at 1200× *g* for 10 min, and the concentration of hemoglobin released in the supernatant was determined by measuring the absorbance at 540 nm. The % hemolysis was calculated as follows: Hemolysis (%) = (Abs_sol_ − Abs_0_)/(Abs_100_ − Abs_0_) × 100, with Abs_sol_ the absorbance measured for the tested condition, Abs_0_ and Abs_100_ the absorbance measured for the negative and positive control, respectively.

## 3. Results

### 3.1. Encapsulation of Terpene Alcohols in Lipid Nanoparticles (LNPs)

Due to their lipophilic character (Figure 1-logP = 4.84 for FAR and 2.9 for GER), the terpene alcohols used cannot be solubilized as a plain aqueous solution at the concentration tested. Thus, these terpene alcohols were encapsulated within LNPs formulated via a phase inversion method [37] to obtain a homogeneous liquid phase containing 28.5 g/L of FAR or GER. This method produced within around 20 min stable monodisperse nanoparticles with a mean hydrodynamic diameter around 60 nm (Table 1). The encapsulation of these terpene alcohols at a mass ratio of 20% had little influence on the formulation process as all particle batches had similar narrow monodisperse size distribution characterized by a low PDI and similar average diameters (Table 2). The size distribution of LNPs was not changed after six months of storage at 4 °C.

### 3.2. Both Terpene Alcohol-Loaded LNP Were Non-Active against Susceptible and MCR-1 E. coli but Enhanced the Effect of Colistin

FAR and GER alone had MIC higher than 2096 mg/L against *E. coli* J-53 and its MCR-1 transconjugant and were thus considered non-ATB. In this condition, FIC index analysis can difficultly be used to evaluate the influence of these terpene alcohols on the colistin efficacy. In addition, the curves showing the variation in colistin MIC as a function of FAR and GER concentrations displayed sigmoidal profiles (Figure 2).

Thus, to quantify the magnitude of the effect of these terpene alcohols on the efficacy of colistin, the data were analyzed using an E_max_ model described by Equation (1). The predicted model parameters, presented in Table 3, were well estimated (relative standard error (RSE) on the mean values were less than 2.9% (see Appendix A)) and most of the observed data were evenly distributed around the mean posterior predictions (line) within the 90% credible intervals (Figure 2).

Hence, this model adequately described the initial steep decrease in colistin MIC observed in the presence of low concentration (<10 mg/L) of FAR-loaded LNP with *E. coli* J53 and its MCR-1 transconjugant. Based on comparable EC_50_ values of around 2 mg/L (Table 3), FAR-loaded LNP potency to decrease the colistin MIC was similar for susceptible and resistant *E. coli*. Yet, the maximal enhancement of the colistin effect (E_max_) induced by the FAR-loaded LNP was around 4 for susceptible *E. coli*, but increased to 16 for the MCR-1 transconjugant, decreasing the colistin MIC from 8 mg/L to 0.5 mg/L.

E_max_ values similar to those attained for FAR-loaded LNP were obtained for the GER-loaded LNP with both *E. coli*. Thus, both terpene alcohols-loaded LNPs were able to restore the colistin MIC against the MCR-1 transconjugant down to 0.5 mg/L (i.e., at a value lower than the susceptibility breakpoints against this bacteria (2 mg/L) [43]). Still, while the potency of both terpene alcohols-loaded LNPs was the same against the colistin-susceptible *E. coli*, the potency of FAR-loaded LNP was higher than that of GER-loaded LNP against the MCR-1 transconjugant, characterized by EC_50_ values of 2.1 mg/L and 35.7 mg/L, respectively.

Farnesol and geraniol-loaded LNPs in combination with colistin at 1/8 of the MIC produced a complete bactericidal effect after 6 h of incubation. Time-kill experiments were performed to evaluate the kinetics of change in bacterial concentrations induced by colistin in the presence of terpene alcohols at concentrations selected from the E_max_ model analysis (Figure 3). Accordingly, the colistin concentration was set at 1/8 of the respective initial MIC value for the two bacteria (0.031 mg/L and 1 mg/L for the *E. coli* colistin susceptible and resistant, respectively), i.e., around MIC/E_max_. Three terpene alcohol concentrations were tested; one corresponding to the base of the sigmoid curves in Figure 2 (10 mg/L for farnesol and 60 mg/L for geraniol), which should give the same effect as pure colistin at its MIC, and two higher terpene alcohol concentrations (30 and 60 mg/L for farnesol and 100 and 200 mg/L for geraniol), which prevented the formation of visible haze in the bacterial suspension in Figure 2 (i.e., maintain at least a bacterial concentration lower than 10^6^ CFU/mL) but could also produce a bactericidal effect.

Colistin alone at a concentration of 1/8 of its MIC (Figure 3, Empty Square) slightly slowed down the growth of both *E. coli* in the first fifth hours, but after 24 h, the bacterial concentrations were similar to those of the antibiotic-free control (empty circle). Similarly, the terpene alcohol-loaded LNPs present at the highest terpene alcohol concentration tested had nearly no effect on the growth of both bacteria and the bacterial concentrations were similar to those of the antibiotic-free control after 24 h. However, for a colistin concentration of 1/8 of its MIC combined to the highest terpene alcohols-loaded LNPs concentrations tested (60 mg/L for FAR and 200 mg/L for GER–solid square), a complete bactericidal effect without regrowth after 30 h for both bacteria was obtained. For colistin-susceptible *E. coli* J53, all terpene alcohols concentrations tested yielded comparable high initial rates of decrease in bacterial concentrations and induced a bactericidal effect (log CFU/mL below 2) after 3 h. The gradual increase in concentrations of both terpene alcohols slowed bacterial regrowth that can be observed after the 3 h, to completely prevent it for FAR and GER concentrations of 60 and 200 mg/L, respectively. For *E. coli* MCR-1, the initial kinetics profiles depended on the concentration and terpene alcohol tested. For the FAR-loaded LNPs, the initial decreases in bacterial concentrations rates were slower than those obtained with the colistin-susceptible *E. coli* and increased with the gradual increased in FAR concentration. For the GER-loaded LNPs, the lowest concentration tested (60 mg/L) was not able to produce a decrease in the bacterial concentration. Increase this concentration to 100 and 200 mg/L gradually increased the rate in bacterial concentrations decrease. For this MCR-1 bacterium, only the highest terpene alcohol concentrations were able to achieve a bactericidal effect after 6 h of incubation.

### 3.3. Farnesol and Geraniol-Loaded LNPs Enhance the Bacterial Membranes Permeabilization Effect of Colistin

The outer membrane (OM) of Gram-negative bacteria (GNB) can be destabilized when enough colistin molecules bind to it. Then, colistin penetrates into the inner membrane (IM), inducing an increase in its permeability, leakage of intracellular contents and bacterial death [44]. To evaluate the effect of colistin in combination with terpene alcohols-loaded LNPs on the stability of bacterial membranes, the rate of propidium iodide (PI) uptake, used as a dye that only penetrates damaged IM, was measured by following the rate of increase of its fluorescence due to its intercalation with the bacterial DNA. This rate of increase in fluorescence was normalized to the highest value obtained in the presence of colistin alone at 32 mg/L and plotted against colistin concentration (Figure 4). Colistin was associated with terpene alcohol-loaded LNPs to obtain either 30 mg/L of FAR (left side of Figure 4) or 90 mg/L of GER (right side of Figure 4). Higher FAR and GER concentrations resulted in a too rapid increase in PI fluorescence to be experimentally observable under good conditions and to maintain the initial kinetic conditions (i.e., linear fluorescence increase over time).

When possible, the magnitude of the effect of FAR and GER on the permeabilization effect of colistin, was quantify by analyzing the variation of the relative PI uptake rates with colistin concentrations using the E_max_ model described by Equation (2) (Figure 4). The predicted parameters, presented in Table 4, were well estimated (RSE on the mean values were less than 1.72% (see Appendix A)) and most of the observed data were evenly distributed around the mean posterior predictions (line) within the 90% credible intervals.

The EC_50_ obtained for pure colistin against colistin-susceptible *E. coli* J53 (0.89 ± 0.11 mg/L) was significantly lower than the value found for *E. coli* MCR-1 (7.38 ± 0.85 mg/L). This result shows that the MCR-1 plasmid induces a reduction in the potency of colistin to destabilize bacterial membranes. As shown by the similar E_min_ values found for all conditions tested, 30 mg/L of FAR or 90 mg/L of GER alone as LNP had no significant effect on the increase in PI uptake rate measured with both *E. coli*. However, supplementing colistin with terpene alcohol-loaded LNPs increased the predicted E_max_ for *E. coli* J53 from 0.69 ± 0.02 for pure colistin to values higher than 6 and higher than 2 in the presence of FAR at 30 mg/L and GER at 90 mg/L, respectively. Therefore, the efficacy of colistin in disrupting colistin-susceptible *E. coli J53* membranes was enhanced by both terpene alcohol-loaded LNPs, but FAR-loaded LNPs appeared more efficient. Against the colistin susceptible *E. coli*, increases in E_max_ were associated with a shift of the EC_50_ from 0.89 ± 0.11 mg/L to values > 10 mg/L for both terpene alcohols. Thus, similar effects were observed for low concentrations in colistin (with or without terpene alcohol) and the additional effect only appeared for colistin concentrations that already produced maximum effect with pure colistin (Figure 4).

Against *E. coli* MCR-1, the E_max_ values observed with pure colistin (0.97 ± 0.03) were increased to 4.25 ± 0.16 with FAR and to a value ≥ 2.5 with GER. Interestingly, FAR-loaded LNPs also decreased the EC_50_ from 7.38 ± 0.85 mg/L to 2.69 ± 0.25 mg/L, whereas GER-loaded LNPs increased the EC_50_ to over 10 mg/L. Therefore, FAR-loaded LNPs enhanced the bacterial inner membrane permeabilization effect of colistin in the presence of colistin concentrations that had little effect alone.

### 3.4. Farnesol and Geraniol-Loaded LNPs Did Not Induced Red Blood Cell Membrane Permeabilization in the Presence of Colistin

Farnesol and geraniol-loaded LNPs have been shown to enhance the permeabilization effect of colistin on bacterial membranes, but permeabilization of the mammalian cell membrane by colistin is a potential adverse effect that could also be enhanced by these terpene alcohol-loaded LNPs. To evaluate this potential effect, hemolytic activity of colistin alone or in the presence of fixed concentration of terpene alcohol-loaded LNPs was measured on human RBCs (Table 5).

## 4. Discussion

In the currently accepted mechanism of action of colistin, the positively charged peptide cycle of colistin first binds to the anionic lipid A of the LPS of Gram-negative bacteria, disrupting the outer layer of their OM. Then, the acyl tail of colistin interacts with the fatty acids of lipid A, causing greater disruption of the OM allowing colistin to access the IM to cause bacterial lysis [45,46,47]. Resistance to polymyxins in Gram-negative bacteria arises mainly through alterations of their lipid A. For example, the *mcr* genes encode phosphoethanolamine transferases that incorporate phosphoethanolamine into the bacteria’s lipid A, changing the colistin target from a negative to a neutral charge and reducing the affinity with which colistin binds to it [12,47]. Consequently, higher colistin concentration is required to produce an effect. Also, the colistin MIC against *E. coli* J53 was 0.25 mg/L, while the one measured for its MCR-1 transconjugant was 8 mg/L. Similarly, the colistin concentration allowing to obtain half of the propidium iodide (PI) maximal fluorescence increase rate (EC_50_), which correlating with the rate of bacterial IM destabilisation, was 0.86 ± 0.08 mg/L for *E. coli* J53 and 7.38 ± 0.85 mg/L for its MCR-1 transconjugant (Table 4). This result shows that the MCR-1 plasmid induces a reduction in the potency of colistin to destabilize both bacterial membranes. A recent study suggested that colistin also binds to LPS in the IM and that this interaction is essential for colistin permeabilisation and bactericidal activity. This study also shows that MCR-1-mediated colistin resistance confers protection against colistin via the presence of modified LPS within the IM, rather than the OM [47].

Acyclic terpene alcohols such as farnesol are known to interact with bacterial OM and IM membranes, changing their physical properties such as fluidity [29,30,35]. For example, farnesol was shown to interfere with *A. baumannii* membrane structure [35]. Therefore, we assumed that farnesol or geraniol-loaded LNP could destabilize Gram-negative bacteria membranes and thus facilitate colistin killing action. At the concentrations tested, the two LNPs loaded with farnesol or geraniol were inactive against susceptible and MCR-1 *E. coli* J53. Indeed, they did not modify the kill-curve profiles compared to the control (Figure 3) nor did they affect the rate of increase in the fluorescence of PI (Figure 4), which correlates with the destabilization of both bacterial membranes. However, LNPs loaded with the two terpene alcohols enhanced the effect of colistin and reduced its MIC against these strains by 4 to 16 times (Figure 2). Additionally, both terpene alcohol-loaded LNPs in combination with colistin at 1/8 of its MIC enhanced the initial bacterial killing rate and produced a complete bactericidal effect after 6 h of incubation for susceptible and MCR-1 *E. coli* J53 in the presence of 60 mg/L and 200 mg/L of farnesol and geraniol, respectively. In these two conditions, no bacteria regrowth was observed (Figure 3). Based on these results, our study shows that farnesol is more efficient in increasing colistin efficacy than geraniol. This effect was principally observed with *E. coli* J53 MCR-1. A previous study has also shown that nerodiol and farnesol accelerated the bactericidal effect of polymyxin B against *E. coli* (ATCC 25922) [31]. The authors suggested that this effect was due to the interaction of these terpene alcohols with the bacterial membranes. Similarly, resveratrol, a hydrophobic polyphenol that also induces membrane instability [48], reduces the MICs of polymyxin B against *Klebsiella pneumoniae* and *E. coli* isolates up to 512 fold at a concentration of 128 μg/mL [49]. Farnesol has also been shown to potentiate the activity of colistin against several strains of *A. baumannii* due to its membrane destabilisation capacity [35]. In this study carried out on *A. baumannii*, farnesol at 111 mg/L combined with colistin at 1 mg/L produced a bactericidal effect without regrowth, whereas bacteria regrew in the presence of pure colistin. Similar results found with different *Enterobacteriaceae* (*E. coli and A. baumannii*) from different laboratories suggest that the colistin potentiation effect of farnesol may work on other bacteria of this family.

To determine whether the inner membrane (IM) integrity was more affected by terpene alcohols-LNPs/colistin combination than pure colistin, we measured the uptake of the inner membrane (IM) impermeable fluorophore propidium iodide (PI). An intact IM prevents entry of PI into the bacteria and the subsequent fluorescence. Therefore, PI uptake represents a quantitative read-out for colistin-mediated IM permeabilisation. Against colistin susceptible *E. coli* J53, similar IM permeabilisation rates than with pure colistin were observed in the presence of terpene alcohols-loaded LNP for concentrations in colistin lower than 5 mg/L (Figure 4-left panels). For the susceptible *E. coli*, an additional increase in PI uptake rates induced by the terpene alcohols only appeared for colistin concentrations (above 5 mg/L) that already produced a maximum effect with pure colistin. This additional IM destabilization effect induced by the terpene alcohols was not observed for colistin and terpene alcohols concentrations that were bactericidal, for example for 0.031 mg/L of colistin and 30 mg/L of farnesol (Figure 3-plain triangle), while colistin alone at 0.031 mg/L had almost no effect (Figure 3–empty square). Hence, the increased destabilization rate of IM observed with the susceptible *E. coli* strain might not be the effect responsible for the increased bactericidal effect in the presence of the terpene alcohol/colistin combination. Yet, IM permeabilization seems essential for colistin-induced cell lysis activity [12,45,47].

Contrariwise, for the MCR-1 *E. coli*, increased IM permeabilization rate was observed for colistin concentrations that correlate with the concentrations producing the bactericidal effect. At 1 mg/L of colistin and 30 mg/L of farnesol, a strong initial bactericidal effect was obtained (Figure 3) and the EC_50_ of colistin that induce half of the maximal PI uptake rate in the presence of 30 mg/L of farnesol was of 2.69 ± 0.25 mg/L (Table 5). Thus, farnesol might enhance colistin bactericidal effect against the MCR-1 strain by improving its IM permeabilization effect. In the same way, it was previously found that MCR-1-mediated colistin resistance confers protection against colistin via the presence of modified LPS within the IM, rather than the OM [47].

Colistin is an antibiotic that was almost abandoned for many years due to its reported toxicity. Hence, increase its efficacy could also allow reducing the dose needed even against susceptible bacteria. Interestingly, the LNP loaded with farnesol improved bacterial membranes destabilization due to colistin, but did not affect the red blood cells membrane integrity.

## 5. Conclusions

In this study, it was shown that 10 mg/L of farnesol formulated as lipid nanoparticles restored the sensitivity of *E. coli* MCR-1 to colistin. Due to this effect, lower concentrations of colistin (eight times lower) were necessary to obtain a bactericidal effect. This effect was due in part to the facilitation of the bacterial membrane permeabilization effect of colistin in the presence of farnesol. These results have important implications for the repositioning of non-antibiotic drugs such as farnesol for antimicrobial purposes, which could accelerate the discovery of new therapies to combat the rapid emergence of colistin resistance.

## 6. Patents

The patent WO2020021052A2 results from the work reported in this manuscript.

## Figures and Tables

**Figure 1 pharmaceutics-13-01849-f001:**
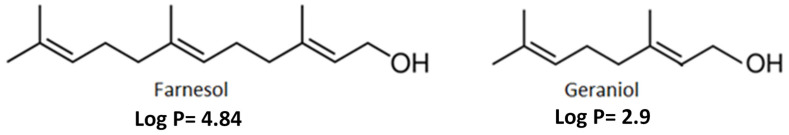
Chemical structure of farnesol and geraniol.

**Figure 2 pharmaceutics-13-01849-f002:**
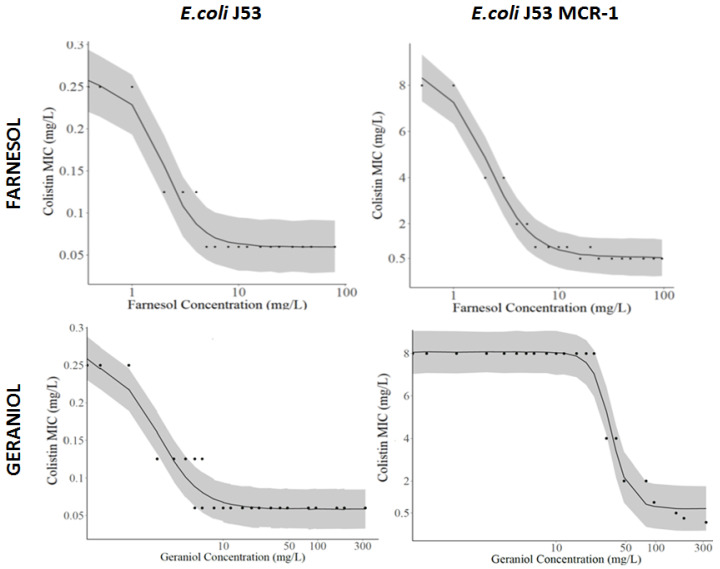
Colistin MIC (mg/L) versus farnesol (**top**) or geraniol (**bottom**) concentrations (mg/L) measured for *E. coli* J53 (**left**) and the corresponding *mcr*-1 transconjugate *E. coli* J53_MCR-1 (**right**). The shaded area are the 90% credible intervals around the mean values (lines) obtained using the E_max_ model (1). (*n* = 3, notice that most of the points are overlayed).

**Figure 3 pharmaceutics-13-01849-f003:**
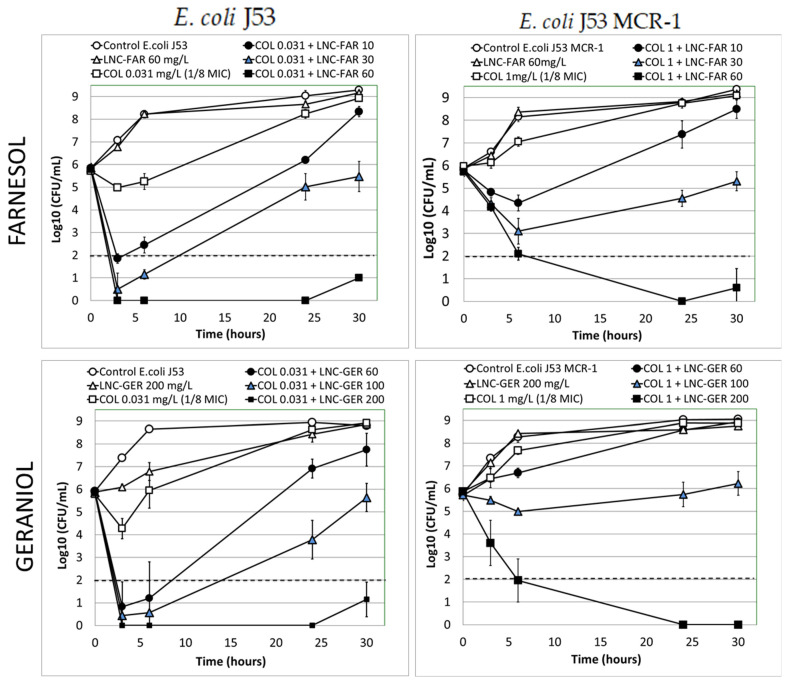
Time-kill curves (*n* = 3–6) obtained for *E. coli* J53 (left panels) and *E. coli* J53_MCR-1 (right panels) in the presence of colistin base alone at a concentration of 1/8 of its MIC (empty square) or supplemented with FAR-loaded LNP at FAR concentrations of 10, 30, and 60 mg/L (Top plots–solid dots) or supplemented with GER-loaded LNP at GER concentrations of 60, 100, and 200 mg/L (bottom plots–solid dots). Terpene alcohol effect controls were made using the terpene alcohol-loaded LNP alone in the presence of the highest terpene concentration tested (empty triangles). The dotted horizontal line represents the limit of quantification of the bacteria concentration.

**Figure 4 pharmaceutics-13-01849-f004:**
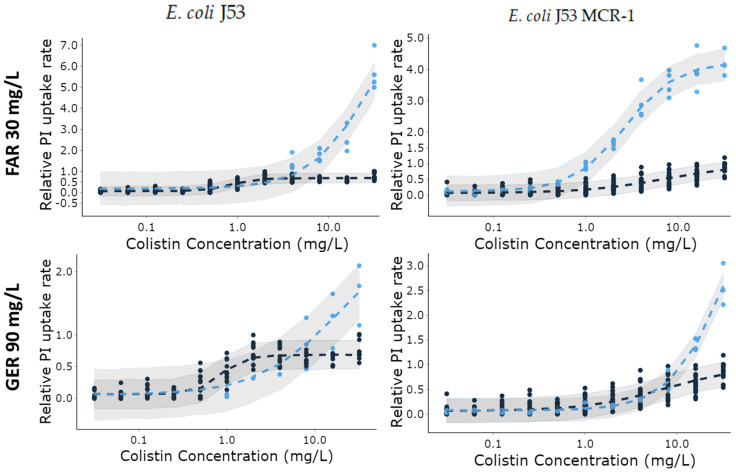
Relative propidium iodate (PI) uptake rate by *E. coli* J53 (left panels) and *E. coli* J53 MCR-1 (right panels) measured as a function of colistin concentration alone (black circles) or in the presence of 30 mg/L of farnesol (top-blue circles) or 90 mg/L of geraniol (bottom-blue circles). Uptake rates were normalized against the highest uptake rate measured in the presence of colistin alone at 32 mg/L. When the data described a complete sigmoidal shape, they were analyzed using the E_max_ equation (Equation (2)). The shaded area are the 90% credible intervals around the mean values (lines). (4 ≤ *n* ≤ 20).

**Table 1 pharmaceutics-13-01849-t001:** Composition of the LNP. Theoretical terpene alcohol loading was calculated as the mass of terpene alcohols over the mass of all the compounds.

Excipients	Mass (mg)
Solutol^®^ HS15	850
Terpene Alcohols	500
Phospholipon^®^	75
Labrafac lipophile WL1349	1028
Theoretical terpene alcohol loading	20.4%

**Table 2 pharmaceutics-13-01849-t002:** Particles size distribution properties of the blank and terpene alcohols-loaded LNPs. PDI (polydispersity index).

	Mean Diameter (nm)	PDI
**Blank LNP**	58.5 ± 2.6	0.074 ± 0.020
**Farnesol LNP**	60.2 ± 1.6	0.052 ± 0.012
**Geraniol LNP**	62.1 ± 1.4	0.071 ± 0.031

**Table 3 pharmaceutics-13-01849-t003:** E_max_ model fitting parameters values ± SE calculated for curves representing the colistin MIC (mg/L) measured for *E. coli* J53 or its MCR-1 transconjugant (*E. coli* J53_MCR-1) versus farnesol or geraniol concentrations (mg/L).

Terpene Alcohol	Strain	MIC_0_	MIC_∞_	E_max_^(1)^	EC_50_^(2)^ (mg/L)	*γ*
Farnesol	*E.coli* J53	0.26 ± 0.01	0.06 ± 0.01	4	2.0 ± 0.3	3.1 ± 2.3
	*E.coli* J53_MCR-1	8.68 ± 0.56	0.54 ± 0.16	16	2.1 ± 0.2	2.1 ± 0.4
Geraniol	*E.coli* J53	0.26 ± 0.01	0.06 ± 0.01	4	2.0 ± 0.2	1.9 ± 0.2
	*E.coli* J53_MCR-1	8.07 ± 0.17	0.59 ± 0.28	16	35.7 ± 2.0	4.8 ± 1.2

^(1)^ E_max_: Maximal enhancement of the colistin effect (colistin MIC_0_/colistin MIC_∞_). ^(2)^ EC_50_: terpene alcohol concentration (mg/L) needed to get half E_max_.

**Table 4 pharmaceutics-13-01849-t004:** E_max_ model fitting parameters values estimated for curves representing PI relative uptake rate measured for *E. coli* J53 or *E. coli* J53_MCR-1 versus colistin alone or associated with 30 mg/L of FAR or 90 mg/L of GER. N.A.: not applicable.

Strain		E_max_	EC_50_ (mg/L)	*γ*	E_min_
*E.coli* J53	Colistin	0.69 ± 0.22	0.89 ± 0.11	3.48 ± 1.62	0.07 ± 0.02
	Colistin + FAR 30 mg/L	>6	>10	N.A.	0.19 ± 0.10
	Colistin + GER 90 mg/L	>2	>10	N.A.	0.05 ± 0.04
*E.coli* J53_MCR-1	Colistin	0.97 ± 0.03	7.38 ± 0.85	1.03 ± 0.01	0.07 ± 0.02
	Colistin + FAR 30 mg/L	4.25 ± 0.16	2.69 ± 0.25	1.56 ± 0.19	0.11 ± 0.07
	Colistin + GER 90 mg/L	>2.5	>10	N.A.	0.08 ± 0.03

**Table 5 pharmaceutics-13-01849-t005:** Hemolytic activity of colistin (% of triton x-100 positive control) for different concentrations of pure colistin solution or in the presence of LNP to obtain 60 mg/L of FAR or 100 mg/L of GER. Values presented are means +/− SD (*n* = 3).

Colistin Concentration (mg/L)		0	5	10	50	100	200
**% Hemolysis**	Colistin	0.00 ± 0.14	−0.16 ± 0.04	−0.27± 0.05	0.00± 0.08	−0.19± 0.04	−0.22± 0.18
	Colistin + FAR 60 mg/L	−0.07 ± 0.16	−0.05 ± 0.08	0.02 ± 0.02	0.16 ± 0.09	0.45 ± 0.02	1.19 ± 0.19
	Colistin + GER 100 mg/L	0.47 ± 0.06	−0.12 ± 0.01	0.06 ± 0.04	−0.07 ± 0.03	−0.02 ± 0.08	0.00 ± 0.04

## Data Availability

The data presented in this study are available on request from the corresponding author.

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
