# Peer review of "Lipid Nanoparticles Loaded with Farnesol or Geraniol to Enhance the Susceptibility of E. coli MCR-1 to Colistin"

_pharmaceutics, 2021, doi:10.3390/pharmaceutics13111849_

Round 1

Reviewer 1 Report

This article deals with the evaluation of the combined activity of colistin with terpene alcohols such as farnesol  and  geraniol against two types of   strains belonging to the species E,coli (susceptible E. coli J53 and resistant transconjugant E.coli J53 mcr 1 containing mobilized colistin resistant  plasmid genes). The colistin (an old antibiotic used in the past) is currently considered a last resort for the MDR Gram-negative but its toxicity limits its use in the routine. The above association with terpene alcohols inserted in water-dispersible LNP (Lipid Nanoparticles in order to bypass their high lipophilicity) results very useful  in combating these dangerous infections as well as in restoring the full activity of colistin at lower dosage then decresing its toxicity.

The paper turns out to be interesting, well written, clearly exposed and significant for establishing  an alternative and quite efficacious therapy against  these fastidious infections. Just a few minor corrections  or clarifications are required.  In line 135 one word is missing  "of the Emax and...... is the slope factor".  In line 145 the inoculum is reported to be  1x106 CFU/mL. Generally in the time Kill studies an inoculum  of 1x105  CFU/mL is used.  Is there any reason for this? And why is it used the concentration of 1/8 of the respective MICs and not the more common concentrations of 1/2 or 1/4 (see line 147)?  Moreover  are  the values of  the  Emax for both Farnesol and Geraniol  ( values 4 and 6, Table 3) equal or should these be different considering the diverse activity of the two above compounds? In the legenda of igure 4 there is a mistake (E. coli J53 (right panels) and E. coli J53 MCR-1 (left panels)  instead of the opposite E. coli J53 (left  panels) and E. coli J53 MCR-1 (right panels). Lastly in lines 385 an 386 the verbs "destabilized and facilitated" should be modified in "destabilize and facilitate"

Author Response

This article deals with the evaluation of the combined activity of colistin with terpene alcohols such as farnesol  and  geraniol against two types of   strains belonging to the species E,coli (susceptible E. coli J53 and resistant transconjugant E.coli J53 mcr 1 containing mobilized colistin resistant  plasmid genes). The colistin (an old antibiotic used in the past) is currently considered a last resort for the MDR Gram-negative but its toxicity limits its use in the routine. The above association with terpene alcohols inserted in water-dispersible LNP (Lipid Nanoparticles in order to bypass their high lipophilicity) results very useful  in combating these dangerous infections as well as in restoring the full activity of colistin at lower dosage then decresing its toxicity.

The paper turns out to be interesting, well written, clearly exposed and significant for establishing an alternative and quite efficacious therapy against  these fastidious infections.

Just a few minor corrections  or clarifications are required.  In line 135 one word is missing  "of the Emax and...... is the slope factor". 

Thank you, we have corrected the typo.

 In line 145 the inoculum is reported to be  1x10CFU/mL. Generally in the time Kill studies an inoculum  of 1x105  CFU/mL is used.  Is there any reason for this?

It is true that time Kill studies can performed with a starting inoculum of 1x105  CFU/mL, but several teams specialised in pharmacology of antimicrobials do so with an inoculum of 1x10CFU/mL (doi:10.1093/jac/dkx328 (Johan W. Mouton lab),  10.1093/jac/dkx537, 10.1093/jac/dkw082 (Hartmut Derendorf lab), 10.1128/AAC.04182-14 (Roger L. Nation lab)). A higher inoculum gives more chances to observe the development of bacteria resistant to the treatment.

 And why is it used the concentration of 1/8 of the respective MICs and not the more common concentrations of 1/2 or 1/4 (see line 147)? 

The MIC of MCR E. coli was 8 mg/L, so 1/8 of this value corresponds to a clinically achievable concentration of colistin in human lungs using the doses of colistin usually used. This has been added to the manuscript in line 147.

 Moreover  are  the values of  the  Emax for both Farnesol and Geraniol  ( values 4 and 6, Table 3) equal or should these be different considering the diverse activity of the two above compounds?

The Emax (maximum increase in colistin effect (colistin MIC0 / colistin MIC∞)) was 16 for both farnesol and geraniol, but geraniol was less potent than farnesol because approximately 15 times more geraniol than farnesol was required to achieve this max effect.

In the legenda of figure 4 there is a mistake (E. coli J53 (right panels) and E. coli J53 MCR-1 (left panels)  instead of the opposite E. coli J53 (left  panels) and E. coli J53 MCR-1 (right panels).

Thank you, we have corrected the typo.

Lastly in lines 385 an 386 the verbs "destabilized and facilitated" should be modified in "destabilize and facilitate"

Thank you, we have corrected the typo

Reviewer 2 Report

The paper is generally well written and merits publication; however, the quality of the paper can be enhanced if the following points can be addressed.

  1. Did the authors study the stability of the drug loaded lipid nanoparticles?
  2. Did the authors check the drug release profile for the drugs that are used in the study? 
  3. How is the loading capacity of the drugs determined?
  4. Can the proposed drug combination work independantly without nanocarrier formulation?

Author Response

The paper is generally well written and merits publication; however, the quality of the paper can be enhanced if the following points can be addressed.

  1. Did the authors study the stability of the drug loaded lipid nanoparticles?

The particle size distribution and efficiency of the nanoparticle suspensions did not change after 6 months of storage in the refrigerator at 4°C. This was mentioned in the manuscript in line 214.

2. Did the authors check the drug release profile for the drugs that are used in the study? 

3. How is the loading capacity of the drugs determined?

We focused the study on the ability of the formulations to improve the efficacy of colistin against E. coli MCR-1. We used the formulations as whole systems, without a purification step, so the initial amount and concentration of terpene alcohol was known, but we did not know what relative fraction was in the lipid nanoparticles or in the aqueous dispersion phase. Because these two compounds are very lipophilic (logP values for farnesol and geraniol are 2.9 and 4.84), we expect that most of the molecules would remain in the oil phase forming the nanoparticles. Moreover, generally, these molecules are assayed by a gas chromatography method that we do not have in the laboratory

4. Can the proposed drug combination work independantly without nanocarrier formulation?

The aqueous solubility of farnesol and geraniol is very low; their respective logP values are 2.9 and 4.84, so it is not possible to solubilize them in water at the concentration tested. Other methods than lipid nanoparticles, such as cosolvents or surfactants, could be used to solubilize them, but they may be more difficult to use in humans.
